# Newtic1 Is a Component of Globular Structures That Accumulate along the Marginal Band of Erythrocytes in the Limb Blastema of Adult Newt, *Cynops pyrrhogaster*

**DOI:** 10.3390/biomedicines10112772

**Published:** 2022-11-01

**Authors:** Xutong Chen, Ryo Ando, Roman Martin Casco-Robles, Martin Miguel Casco-Robles, Fumiaki Maruo, Shuichi Obata, Chikafumi Chiba

**Affiliations:** 1Graduate School of Life and Environmental Sciences, University of Tsukuba, Tennodai 1-1-1, Tsukuba 305-8572, Ibaraki, Japan; 2Graduate School of Science and Technology, University of Tsukuba, Tennodai 1-1-1, Tsukuba 305-8572, Ibaraki, Japan; 3Faculty of Life and Environmental Sciences, University of Tsukuba, Tennodai 1-1-1, Tsukuba 305-8572, Ibaraki, Japan; 4Department of Anatomical Sciences, Faculty of Allied Health Sciences, Kitasato University, Kitasato 1-15-1, Minami-ku, Sagamihara 252-0373, Kanagawa, Japan; 5Regenerative Medicine and Cell Design Research Facility, Faculty of Allied Health Sciences, Kitasato University, Kitasato 1-15-1, Minami-ku, Sagamihara 252-0373, Kanagawa, Japan; 6Department of Histology and Cell Biology, Faculty of Medicine, Yokohama City University, Fuku-ura 3-9, Kanazawa-ku, Yokohama 236-0004, Kanagawa, Japan

**Keywords:** Newtic1, TGFβ1, BMP2, microtubule, erythrocyte, polychromatic normoblasts (PcNobs), orthochromatic normoblasts (OcNob), erythrocyte clump (EryC), limb regeneration, newt

## Abstract

In adult newts, when a limb is amputated, a mesenchymal cell mass called the blastema is formed on the stump, where blood vessels filled with premature erythrocytes, named polychromatic normoblasts (PcNobs), elongate. We previously demonstrated that PcNobs in the blastema express an orphan gene, *Newtic1*, and that they secrete growth factors such as BMP2 and TGFβ1 into the surrounding tissues. However, the relationship between Newtic1 expression and growth factor secretion was not clear since *Newtic1* was thought to encode a membrane protein. In this study, we addressed this issue using morphological techniques and found that the Newtic1 protein is a component of globular structures that accumulate at the marginal band in the cytoplasm along the equator of PcNobs. Newtic1-positive (Newtic1(+)) globular structures along the equator were found only in PcNobs with a well-developed marginal band in the blastema. Newtic1(+) globular structures were associated with microtubules and potentially incorporated TGFβ1. Based on these observations, we propose a hypothesis that the Newtic1 protein localizes to the membrane of secretory vesicles that primarily carry TGFβ1 and binds to microtubules, thereby tethering secretory vesicles to microtubules and transporting them to the cell periphery as the marginal band develops.

## 1. Introduction

To varying degrees, four-limbed vertebrates (tetrapods), including humans, generally have the ability to regenerate lost complex tissues or body parts after trauma early in development, but as they grow and become adults, this ability is reduced or lost, and the deficient areas heal instead by being covered with fibrotic tissue [1,2]. Contrary to this general rule, newts, which belong to a group of the family Salamandridae in urodele amphibians, have the ability to repeatedly regenerate lost body parts, regardless of their age, even after reaching adulthood beyond metamorphosis [3,4,5,6,7,8,9,10,11,12,13,14]. It is believed that this outstanding regenerative ability of adult newts is based on a mechanism of cellular reprogramming/dedifferentiation that is unique to newts: in adult newts, terminally differentiated somatic cells, which have already lost premature traits such as multipotency factor expression and proliferative activity and have become highly specialized for specific physiological functions, are reprogrammed/dedifferentiate into stem/progenitor-like cells upon trauma. For example, retinal pigment epithelium (RPE) cells in the eyes are reprogrammed into RPE stem cells for retinal regeneration [1,6,7,9] while striated muscle fibers (tubular, multinucleated cells with sarcomeres) in the limbs dedifferentiate into mononucleated myogenic cells for muscle regeneration [10,14]. This ability could be interpreted as allowing adult newts to mobilize stem/progenitor-like cells from terminally differentiated somatic cells as regenerative material to complement resident somatic stem/progenitor cells that have reduced their contribution to regeneration as adults [14]. Therefore, body part regeneration in adult newts is a useful model system for basic research toward new regenerative medicine combining stem cells and dedifferentiation. We have been studying the mechanism of the regeneration of body parts using adult fire-bellied newts, *Cynops pyrrhogaster*.

In a previous study, using an adult *C. pyrrhogaster* limb regeneration system, we comprehensively searched for unique genes whose number of transcripts was significantly elevated in the blastema, a cell mass that appears at the tip of an amputated limb, and discovered an orphan gene, *Newtic1*, which is found only in urodele amphibians [15]. That study [15] revealed the following. *Newtic1* theoretically encodes a 40.7 kD protein that has a transmembrane domain at the N-terminus followed by a cytoplasmic domain. The Newtic1 protein is expressed on a subset of erythrocytes, namely polychromatic normoblasts (PcNobs), premature erythrocytes that make up 80–90% of circulating erythrocytes. PcNobs are transparent or light pink in color, suggesting that hemoglobin synthesis has not started or is in progress. Newtic1 protein is localized along the equator of PcNobs. It must be noted that in adult newts, normoblasts are produced in the spleen and mature in blood vessels during circulation and that fully mature erythrocytes in adult newts are red due to their large amount of hemoglobin but still retain nuclei, corresponding to human orthochromatic normoblasts (OcNob). OcNobs of adult newts never express Newtic1 protein.

Newtic1-positive (+) PcNobs are observed in at least two situations [15]. The first is in normal circulating blood, in which only about 20% of PcNobs express Newtic1 protein, which gathers around monocytes, forming erythrocyte clump (EryC)-monocyte complexes. The minimum size of an EryC-monocyte complex is comprised of 1–2 monocytes and 6–10 PcNobs. The second is in regenerating limbs, as predicted at the time of *Newtic1* gene isolation. At Stage I (5 post-operative days (pod)), when the amputation surface of the limb is covered by the wound epidermis and lymphocytes, including monocytes, accumulate between the amputation surface and the wound epidermis, PcNobs retained in the vessels dilated by the inflammatory reaction just below the amputation surface begin to express Newtic1 protein. At Stage II (14 pod), when the blastema begins to protrude, emerging Newtic1(+) PcNobs enter the blastema along the inside of extending vessels, and their number and expression of the Newtic1 protein increases. As the blastema grows further and reaches Stage III (27 pod), when the entire surface of the blastema is covered by pigmented epithelium and capillaries extending beneath the epidermis, the density of Newtic1(+) PcNobs increases in the distal part of the blastema. Later, as the cartilage begins to differentiate in the regenerating limb, elongated blood vessels fuse to form loops, and as circulation resumes, Newtic1(+) PcNobs are cleared by normal peripheral blood.

We further showed in our previous study that erythrocytes of adult newts contain transcripts of numerous secreted factors, including growth factors (TGFβ1, IGFII, BMP2, PDGFC, VEGEC, and nsCCN) and matrix metalloproteinases (Col-a, Col-b, MMP3/10, MMP9, and MMP21) [15]. We also demonstrated by immunocytochemistry that TGFβ1 and BMP2 are present in erythrocytes at the protein level. Focusing on Newtic1(+) PcNobs in the growing blastema, we further demonstrated that these cells lost immunoreactivity to TGFβ1 and BMP2 from the cytoplasm when they were translocated into the blastema, suggesting the possibility of growth factor secretion by Newtic1(+) PcNobs during blastema growth. However, it remains elusive how the expression of Newtic1, a putative membrane protein, is linked to the secretion of growth factors from PcNobs.

In this study, we addressed the above issue using morphological techniques as a basis for investigating the physiological roles of Newtic1 and Newtic1(+)PcNobs in limb regeneration of adult newts. We found that the Newtic1 protein localizes to cytoplasmic globular structures, which accumulate along the marginal band of PcNob in the limb blastema. Our observations provide insights leading to a hypothesis that Newtic1 protein might be a component of membrane vesicles containing growth factors such as TGFβ1 and may contribute to the transport of these factors by tethering membrane vesicles to tubulin fibers that become aligned just below the equatorial plane of PcNob during blastema formation.

## 2. Materials and Methods

Experiments presented herein were performed at Kitasato University, Yokohama City University, and the University of Tsukuba. All methods were carried out in accordance with the ARRIVE guidelines as well as the Regulations for the Handling of Animal Experiments in each university. Experiments using live animals were conducted only at the University of Tsukuba. All experimental protocols for live animals were approved by the Animal Care and Use Committee of the University of Tsukuba (170110).

### 2.1. Animals

Adult Japanese *C. pyrrhogaster* (total body length: male, about 9 cm; female, 11–12 cm) was used. Newts were captured from Aichi, Fukui, Fukushima, and Hyogo prefectures by a supplier (Aqua Grace, Yokohama, Japan) and stored at the University of Tsukuba. A strain of *C. pyrrhogaster*, Toride-Imori, was also used [16]. The animals were reared at 18–22 °C under natural light conditions until experiments as described previously [12,13,14,15].

### 2.2. Anesthesia

Animals were anesthetized in 0.1% FA100 (4-allyl-2-methoxyphenol; LF28C054; DS Pharma Animal Health, Osaka, Japan) dissolved in water at room temperature (RT: 22 °C) for 45 min, as described previously [12,13,14,15].

### 2.3. Limb Amputation

After anesthesia, animals were rinsed in Elix water (Merk Millipore, Sigma-Aldrich, Tokyo, Japan) and dried on a paper towel (Elleair Prowipe, Soft High Towel, Unbleached, 4P; Dio Paper Corporation, Tokyo, Japan). After tying a tourniquet—a twisted string made of a piece of soft paper (Elleair Prowipe, Soft Wiper S200; Dio Paper Corporation)—just below the shoulder on the right forelimb, the forearm was amputated in the middle (at the mid-zeugopod region) by a razor blade (FA-10; FEATHER Safety Razor Co., Ltd., Osaka, Japan) as described previously [12,14]. Amputees were momentarily placed on dry paper towels until the bleeding stopped. After the tourniquet was removed, they were transferred to moist containers with a lid containing air vents (up to three newts per container of length 200 mm × width 150 mm × height 55 mm) and allowed to recover. The moist container was always kept in a semi-dry condition in which the bottom was covered by a moist paper towel (Elleair Prowipe, Soft High Towel, Unbleached, 4P) that was tightly wrung [12,13]. Paper towels were replaced with new ones every other day. The stages of limb regeneration were determined according to previous criteria [15].

### 2.4. Tissue Preparation

For immunohistochemistry of limb blastema, forelimbs were re-amputated in the middle of the upper arm under anesthesia and fixed in a modified Zambony’s fixative, which was prepared by dissolving 0.2% picric acid in 2% paraformaldehyde (PFA)-containing phosphate-buffered saline solution (PBS; pH 7.5), at 4 °C for 6 h [15]. Limb samples were washed thoroughly with PBS at 4 °C (5 min × 2, 10 min × 2, 15 min × 2, 30 min × 2, and 1 h × 2) and then allowed to soak in 30% sucrose in PBS at 4 °C. They were embedded into Tissue-Tek O.C.T. Compound (4583; Sakura Finetek USA, Inc., Torrance, CA, USA), frozen at about −30 °C in a cryostat (CM1860; Leica Biosystems, Tokyo, Japan) and sectioned at about 20 μm thickness. Tissue sections were attached to gelatin-coated coverslips (18 × 18 mm^2^, Thickness No. 1; Matsunami, Osaka, Japan), air-dried, and stored at −20 °C until use. After blastemal samples were collected, animals were kept in their original moist containers and allowed to regenerate for use in other studies.

### 2.5. Blood Cell Preparation

Normal circulating blood (named ‘intact blood’ hereafter) was collected from intact limbs immediately after amputation. Blastemal blood was collected from regenerating limbs at 3–4 weeks post amputation by making a small slit on the top of the blastema with the tip of an FA-10 razor blade (see Figure 1D). Blood was harvested from 2–5 animals in 2 mL PBS (pH 7.5) in a plastic dish (inner diameter: 35 mm; PS-30; MonotaRO Co., Ltd., Hyogo, Japan) on ice. About 80–100 µL of intact blood and about 10–15 µL of blastemal blood could be collected per animal. As for the blastemal blood, further collection should be avoided in order to minimize contamination of intact blood. About half of those volumes were blood cell components, which were collected into one 1.5 mL protein low-binding tube (Proten LoBind Tube, 022431081; Eppendorf AG, Hamburg, Germany) by two centrifugations (80 g, 1 min), and rinsing with 1 mL of chilled PBS, followed by centrifugation (80 g, 1 min). These cell samples were used for the following experiments. Developmental stages of erythrocytes were determined according to previous criteria [15]. After the collection of blood samples, animals were kept in their original moist containers and allowed to regenerate for use in other studies.

### 2.6. Antibodies

Antibodies used for immunostaining are listed in Appendix A.

### 2.7. Immunostaining of Tissue Sections and Blood Cells

Immunofluorescence staining of tissue sections was carried out as described previously [15]. Briefly, tissue sections were washed thoroughly (PBS, 0.2% TritonX-100 in PBS, PBS; 15 min each), incubated in blocking solution (5% bovine serum albumin (BSA, 050 M 1599; Sigma-Aldrich in Merck, Tokyo, Japan)/2% normal goat serum (S-1000; Vector Laboratories, Burlingame, CA, USA)/0.2% TritonX-100 in PBS) for 2 h, washed as previously indicated, then incubated in primary antibody diluted with blocking solution at 4 °C for 15 h. After washing thoroughly, sections were incubated in Alexa Fluor 488- or rhodamine-conjugated secondary antibody diluted with blocking solution for 4 h and washed thoroughly. For double staining of TGFβ1 and Newtic1, or BMP2 and Newtic1, whose antibodies were produced in rabbits, tissue sections were first stained with primary antibody for TGFβ or BMP2 and secondary antibody conjugated with Alexa Fluor 488 and then stained with primary antibody for Newtic1 and secondary antibody conjugated with rhodamine. After the stained tissue sections were washed, the nuclei of cells were counterstained with DAPI (1:50,000, D1306; Thermo Fisher Scientific, Tokyo, Japan) or TO-PRO-3 Iodide (1:50,000, T3605; Thermo Fisher Scientific). Sections were mounted on a glass slide with 90% glycerol in PBS.

For immunostaining of blood cells, at the initial stage of the study (Figure 1E–H), the collected cells were re-suspended with 1 mL of 4% PFA in PBS (pH = 7.4; RT) and fixed at RT for 2 h. The cells were washed and immunostained in the same way as was performed with tissue sections, except for the use of 1.5 mL tubes and centrifugation for washing. In all other experiments presented in this paper, the collected cells were re-suspended in 1 mL of chilled newt normal saline (115 mM NaCl, 3.7 mM KCl, 3 mM CaCl_2_, 1 mM MgCl_2_, 18 mM D-glucose, 5 mM HEPES; pH = 7.5 [4]), and then 170–190 µL of cell suspension was plated on each of the poly-d-lysine coated 12 mm round coverslips (354086, Corning BioCoat, Poly-d-Lysine Cellware; Discover Laboratory Inc., Bedford, MA, USA) placed in 35 mm plastic dishes (one coverslip per dish). For immunostaining, after the cells were allowed to attach to the cover slip for 20 min at RT, the dishes were carefully filled with 4% PFA in PBS (pH = 7.4; RT), and the cells were fixed for 30 min at RT. Then, the coverslips on which the cells were immobilized were carefully transferred into other 35 mm plastic dishes filled with fresh 4% PFA in PBS (one coverslip per dish), and the cells were fixed for an additional 1 h at RT. After fixation, the coverslips with cells were transferred into a plastic cup (φ86 × 40 mm) containing 50 mL PBS (up to two coverslips per dish) and washed thoroughly (PBS, 0.2% TritonX-100 in PBS, PBS; 15 min each). The cells on coverslips were immunostained in the same way as was performed with tissue sections, except for the process following the use of a gold nanoparticle-conjugated secondary antibody (Goat anti-Rabbit IgG (H&L) Ultra Small; AURION Immuno Gold Reagents & Accessories, Wageningen, The Netherlands) for Newtic1 immunoelectron microscopy.

For Newtic1 immunoelectron microscopy, after the secondary antibody was thoroughly washed, cells were fixed with 2% glutaraldehyde in PBS for 30 min at RT. After washing in PBS (15 min × 3), cells were fixed with 1% OsO_4_ in PBS for 30 min on ice. After washing in chilled MilliQ water (10 min × 6; Merk Millipore), the cells were incubated with R-Gent SE-LM silver enhancement reagent (500011; AURION Immuno Gold Reagents & Accessories) for 22 min at RT and washed in chilled MilliQ water (10 min × 6). The incubation time for silver enhancement was determined by observing reactions under a microscope (BX50; Olympus, Tokyo, Japan). The stained blood cell samples were stored in MilliQ water at 4 °C.

### 2.8. Preparation of Ultrathin Sections of PcNobs for Transmission Electron Microscopy (TEM)

In experiments to examine the presence of an endomembrane system in PcNobs, blood cells attached to coverslips were fixed with 2% PFA/2.5% glutaraldehyde in PBS for 30 min × 2 at RT in the same manner as was conducted with 4% PFA in PBS. After washing in PBS for 5 min × 5 at RT, the cells were fixed with 1% OsO_4_ in PBS for 1 h on ice. The cells were thoroughly washed in chilled PBS for 3 min × 3, followed by washing in chilled MilliQ water for 15 min × 3. The fixed blood cell samples were stored in MilliQ water at 4 °C.

The blood cell samples for immunoelectron microscopy and for observation of an endomembrane system were en bloc stained with 1% uranyl acetate in 50% ethanol overnight at RT, then washed in distilled water. Then, the specimens were dehydrated with a graded series of ethanol (50, 60, 70, 80, 90, 95, and 99.5% for 5 min each and 3 times in 100% for 10 min each) and propylene oxide (twice for 10 min each), then followed by infiltration and embedding in Epon 812 (TAAB Laboratories Equipment Ltd., Berks, UK). Prior to making ultrathin sections, semithin sections (300–500 nm thick) were prepared from the epoxy block containing the sample with an ultramicrotome (ULTRACUT R, Leica Biosystems), stained with toluidine blue, and examined for Newtic1(+) PcNobs under an optical microscope. The epoxy block, whose semithin sections confirmed the existence of Newtic1(+) PcNobs, was further sliced into ultrathin sections (80–90 nm thick) by an ultramicrotome equipped with a diamond knife (Diatome Ltd., Nidau, Switzerland). Sections were doubly stained with uranyl acetate and lead citrate and observed under a transmission electron microscope (JEM-1400 flash, JEOL, Tokyo, Japan) operated at 80 kV.

### 2.9. Organelle Staining in Live Blood Cells

The Golgi apparatus and endoplasmic reticulum (ER) of live blood cells were visualized using the Organelle-ID RGB III Assay Kit (ENZ-51032-K100; Enzo Life Sciences Inc., Farmingdale, NY, USA) according to the manufacturer’s instructions. Briefly, blood cell samples collected in a protein low-binding 1.5 mL tube, as mentioned above, were immediately re-suspended in 500 µL 1× Organelle-ID Reagent III and incubated for 30 min on ice. After the reagent was removed from the tube following centrifugation (80 g, 1 min), the cells were re-suspended in 1 mL of 80% Leibovitz’s L-15 medium (41300-039; Gibco, Thermo Fisher Scientific) containing 10% fetal bovine serum (FBS; Lot#: AJF10577; Cat#: SH30071.03; Hyclone, Cytiva, Tokyo, Japan), and incubated for 60 min at 25 °C in an incubator (HB-100; TAITEC, Saitama, Japan). After the medium was removed from the tube following centrifugation (80 g, 1 min), the cells were re-suspended in 500 µL of a 1× Assay solution in the kit. After the solution was removed from the tube following centrifugation (80 g, 1 min), the cells were re-suspended with 400 µL of the 1× Assay solution. For imaging, the cell suspension was transferred to glass bottom dishes (35 mm Glass Base Dish; IWAKI, AGC TECHNO GLASS Co., Ltd., Shizuoka, Japan) or placed on glass slides (about 30 µL per slide) and overlaid with coverslips (18 × 18 mm^2^, Thickness No.1; Matsunami).

The lysosomes and mitochondria of live blood cells were visualized using the Organelle-ID-RGB I KIT (ENZ-53007-C200; Enzo Life Sciences Inc.) according to the manufacturer’s instructions. Processing of blood cell samples was basically the same as for the Organelle-ID RGB III Assay Kit described above, except that cells were incubated in 0.2% Reagent I diluted in 1 mL of 80% L-15 medium containing 10% FBS for 30 min at 25 °C, and then washed in PBS.

### 2.10. Immunoblotting

Blood cell samples collected from five animals in a protein low-binding 1.5 mL tube, as mentioned above, were further washed three times with 1 mL of chilled PBS via centrifugation (80 g, 1 min) and then used for immunoblotting. After centrifugation and removal of PBS, cells were dissolved in 500 µL cell lysis buffer (0.1% TritonX-100 in PBS containing 1% protease inhibitor cocktail (P8340; Merck Sigma-Aldrich, Tokyo, Japan)) by tapping. After spinning down the content (7740 g, 30 s), the supernatant containing soluble cytoplasmic and plasma membrane proteins was transferred into a fresh protein low-binding 1.5 mL tube (sample 1) and kept on ice. The deposit on the bottom of the tube, which contained nuclei and cytoskeletons, was washed with 500 µL of cell lysis buffer and then with 1 mL of PBS by tapping and spinning down. After PBS was removed, the deposit was mixed with DNase I solution (RQ1 RNase-Free DNase, M6101; Promega, Madison, WI, USA) and incubated for 20 min at 37 °C to degrade DNA. After washing three times in 1 mL PBS by tapping and spinning down, the pellet of cytoskeletal proteins was kept on ice (sample 2).

Prior to denaturing proteins by heat, 100 µL of sample 1 was transferred to a fresh protein low-binding 1.5 mL tube and mixed with 100 µL of 2× Laemmli Sample Buffer (161-0737; Bio-Rad, Hercules, CA, USA) containing 10% 2-mercaptoethanol. For sample 2, the pellet in the tube was immersed in 100 µL of 1× Laemmli Sample Buffer (diluted with PBS) containing 10% 2-mercaptoethanol and dissociated by sonication (46 kH, VS-70U; Iuchi/AS ONE, Osaka, Japan) for a few min. Subsequently, the proteins in the tubes were heat denatured by placing them in boiling water for 5 min. These protein samples were stored at −80 °C until use. Immediately before use, protein samples were thawed, heat denatured, spun down, and placed at RT.

Proteins in samples 1 and 2 (each 20 μL/well) and those of the molecular weight marker (All Blue Prestained Protein Standards, 1610373; Bio-Rad; 10 μL/well) were separated on a 4–15% gradient gel (4561083, Mini-PROTEAN TGX Precast Gels; Bio-Rad) by SDS-PAGE, and transferred to an activated Immun-Blot^®^ PVDF membrane (1620–174; Bio-Rad). The membrane was cut into strips, including lanes for the protein sample of interest and the molecular weight marker. Membrane strips were individually washed in TBST (100 mM Tris-HCl (pH 7.4), 150 mM NaCl, 0.05% Tween20) for 10 min, incubated in blocking solution (5% bovine serum albumin and 2% goat normal serum in TBST) containing 2% avidin D (Avidin/Biotin Blocking kit, SP-2001; Vector Laboratories) for 1 h, washed in TBST (1, 10, 20 min), and then incubated with primary antibody diluted in blocking solution containing 2% biotin (Avidin/Biotin Blocking kit) at 4 °C for 15 h. After washing in TBST thoroughly (1 min, 15 min × 3), they were incubated with biotinylated secondary antibody diluted in blocking solution for 90 min at RT. After a thorough wash in TBST, they were incubated in AB complex (Vectastain ABC Elite kit, PK-6100; Vector Laboratories) prepared with TBST for 90 min at RT. After washing thoroughly in TBST, they were incubated in DAB (SK-4100; Vector Laboratories) for up to 4 h at RT. Once protein bands became visible, they were washed in distilled water (5 min × 6) to stop the reaction.

### 2.11. Image Acquisition and Analysis

Fluorescence images of blood cell samples at the initial stage of the study (Appendix A) were acquired by a charge-coupled device (CCD) camera system (DP73; cellSens Standard 1.6; Olympus) attached to a fluorescence microscope (BX50; Olympus) as described previously [15]. In all other experiments presented in this paper, transmitted light and fluorescence images of blood cell samples were acquired by an all-in-one fluorescence inverted microscope system (BZ-X800; KEYENCE, Osaka, Japan) with filter sets for TRITC (OP-87764; exciter: 545/25 nm; emitter: 605/70 nm), GFP (OP-87763; exciter: 470/40 nm; emitter: 525/50 nm) and DAPI (OP-87762; exciter: 360/40 nm; emitter: 460/50 nm).

Confocal images of tissue sections and blood cells at the initial stage of the study (Figure 1I, Appendix A) were acquired through an LSM510 microscope system (LSM 5.0 Image Browser software; Carl Zeiss, Jena, Germany) as described previously [15]. In all other experiments presented in this paper, confocal images were acquired through a LSM700 microscope system (ZEN 2009, ver. 6.0.0.303; Carl Zeiss, Oberkochen, Germany) with filter sets for rhodamine (Diode 555 Laser; emitter: BP 575–640 nm), Alexa Fluor 488 (Diode 488 Laser; emitter: BP 515–565 nm) and DAPI (Diode 405-5 Laser; emitter: BP 445/50 nm). In experiments to analyze the size of immunofluorescent dots and their positional relationships at a limitation of optical microscope resolution, a 100× oil objective lens was used in combination with a 10× digital zoom. An averaging function for multiple images at the same optical plane (optical section: 300 nm) was also applied to minimize noise, except for taking serial images along the Z-axis.

TEM images were acquired with a digital camera, Phurona, controlled with the software of RADIUS (EMSIS GmbH, Munich, Germany).

Microscope images were analyzed by software for image acquisition systems and by Adobe Photoshop 2022 (San Jose, CA, USA). The area size of immunofluorescence was measured using NIH ImageJ 1.53t (https://imagej.nih.gov/ij/ (accessed on 29 September 2022)). Images of immunoblotted membranes were obtained by a scanner (TS8430; Canon, Tokyo, Japan). Figures were prepared using Adobe Photoshop 2022. Image brightness, contrast, and sharpness were adjusted according to the journal’s guidelines.

### 2.12. Statistics

In all experiments, we obtained data from three or more rounds of independent trials. Statistical analysis was made using BellCurve for Excel (version 3.23 Social Survey Research Information, Tokyo, Japan). Data in the text are presented as the mean ± SE.

## 3. Results

### 3.1. Newtic1 Protein Is Localized in Globular Structures in the Cytoplasm That Accumulate along the Marginal Band of PcNobs

#### 3.1.1. Immunofluorescence Labeling of Newtic1 Protein Revealed the Accumulation of Numerous Fluorescent Dots along the Equator of PcNobs

We collected blood from the apex of the blastema between 3 and 4 weeks after limb amputation (i.e., the transitional period between Stages II and III [15]) (Figure 1A–D). The percentage of Newtic1(+) PcNobs, except for those in EryC-monocyte complexes, in total normoblasts collected from the blastema (i.e., blastemal blood) was significantly higher than that in total normoblasts collected from the limb immediately after amputation (i.e., intact blood): intact blood, 0–7% (2.3 ± 1.1%), *n* = 6; blastemal blood, 4–29% (18.5 ± 4.6%), *n* = 5; Welch’s *t*-test, *p* = 0.02. Using the blastemal blood, we reexamined the localization of Newtic1-immunoreactivity in PcNobs by optical microscopy. Consistent with the findings of a previous study [15], we observed Newtic1-immunoreactivity, by fluorescence, along the equator of PcNobs (Figure 1E). It gave the appearance of a belt along the equator. However, observation under higher magnification revealed that the belt was composed of numerous fluorescent dots (Figure 1F,G). In some of the Newtic1(+) PcNobs, immunoreactive dots were distributed in the cytoplasm in a dispersed manner but also accumulated along the equator (Figure 1H). Confocal microscopy showed that the cores of the fluorescent dots typically had a diameter of 200–300 nm (Figure 1I).

**Figure 1 biomedicines-10-02772-f001:**
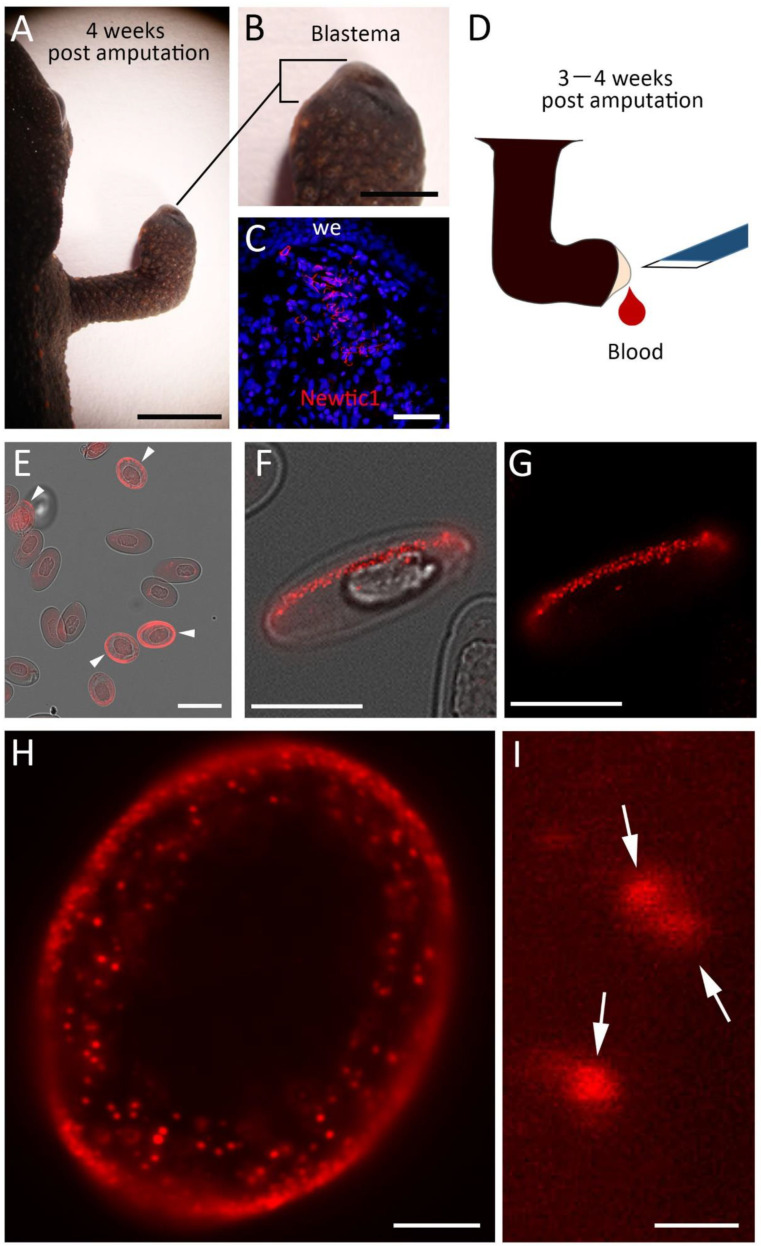
Immunofluorescence labeling of Newtic1 in polychromatic normoblasts (PcNobs). (**A**,**B**) Limb blastema at 4 weeks post amputation. The forelimb was amputated at the midpoint between the elbow and wrist. (**C**) Confocal image of a blastemal section. PcNobs with Newtic1 immunofluorescence (red) along the equator, i.e., Newtic1(+) PcNobs, had accumulated in the blastema. Blue: TO-PRO-3 nuclear stain. we: wound epidermis. (**D**) Collection of blastemal blood. A small slit was made on the top of the blastema using the tip of the blade. (**E**) Cells in blastemal blood. Arrowheads indicate Newtic1(+) PcNobs. (**F**,**G**) Diagonal side view of a Newtic1(+) PcNob. Fluorescent ring along the equator was composed of small dots with immunoreactivity. (**H**) Magnified view of Newtic1(+) PcNob with a 100× objective lens. In this cell, fluorescent dots were distributed not only just beneath the plasma membrane along the equator but also in the cytoplasm. This cell is a relatively young PcNob compared to typical PcNobs such as that in (**F**), because the nucleus was roundish-oval in shape, and the cytoplasmic area was still narrow. (**I**) Confocal image of fluorescent dots with a 100× objective lens. Optical section: 300 nm. Arrows indicate typical fluorescent dots with a core of 200–300 nm in diameter. Scale bars: 5 mm (**A**); 2 mm (**B**); 100 μm (**C**); 40 μm (**E**); 20 μm (**F**,**G**); 5 μm (**H**); 500 nm (**I**).

#### 3.1.2. PcNobs with Well-Developed Marginal Bands Are Likely to Accumulate Newtic1(+) Dots along the Equator

Nucleated erythrocytes in vertebrates, including those of mammalian fetuses, characteristically have a ring-like structure referred to as the marginal band, which is composed of bundles of microtubules that are located just below the cell membrane along the equator [17,18,19]. Therefore, we investigated the relationship between the accumulation of Newtic1(+) dots and the marginal band. We first performed double staining of Newtic1 with α-tubulin, a component of microtubules. Since the host animals for available antibodies were rabbits only, we used α-tubulin antibody for the first stain and Newtic1 antibody for the second stain, and to reduce cross-reactivity of secondary antibodies (i.e., free receptors of the secondary antibody in the first stain react with Newtic1 antibody in the second stain, or the secondary antibody in the second stain reacts with α-tubulin antibody in the first stain), we used IgG Fab fragments as the secondary antibody against α-tubulin antibody and examined various concentrations of α-tubulin antibody. At a dilution rate of 1:100 for α-tubulin antibody, almost all marginal bands of normoblasts were stained by α-tubulin antibody at various intensities (Figure 2A). Under these conditions, the same pattern of reaction was observed in the second stain with the Newtic1 antibody (Figure 2B). The same result was obtained in the second stain with red fluorescent protein (RFP) antibody (Control), suggesting that the secondary antibody in the second stain cross-reacted with α-tubulin antibody (Figure 2C,D). On the other hand, when the concentration of α-tubulin antibody was reduced to 1:1000, only normoblasts with well-developed marginal bands were selectively stained by the α-tubulin antibody (Figure 2E). Unfortunately, even under these conditions, secondary antibody cross-reactivity was unavoidable, and the same staining pattern was observed with the Newtic1 antibody (Figure 2F) and RFP antibody (Control) (Figure 2G,H). However, importantly, normoblasts whose marginal bands were weakly or not at all stained by α-tubulin antibody (1:1000) showed no Newtic1 antibody staining at their equator (three independent trials). This indicates that Newtic1(+) PcNobs (i.e., normoblasts expressing Newtic1 protein along the equator) are among the normoblasts with well-developed marginal bands. In other words, Newtic1(+) PcNobs have well-developed marginal bands. Note that in normoblasts with well-developed marginal bands, the intensity of cytoplasmic staining by α-tubulin antibody tended to be lower relative to the staining along the marginal bands (Figure 2A,C,E,G). This may be due to the polymerization of cytoplasmic-free α-tubulin into microtubules during the development of marginal bands.

#### 3.1.3. Newtic1(+) Dots Are Closely Associated with Microtubules in the Marginal Band

To avoid cross-reactivity with secondary antibodies, we stained the marginal band with a mouse anti-acetylated tubulin antibody, which only stains acetylated α-tubulin within the microtubules (Figure 3A–D). Acetylation of α-tubulin is known to be involved in microtubule dynamics [20,21]. Similar to α-tubulin antibody, acetylated tubulin antibody also stained the marginal bands of almost all normoblasts with varying intensities, although cytoplasmic-free α-tubulin was not stained (Figure 3A). Some of those normoblasts expressed Newtic1 protein along the equator (Figure 3B). Based on a comparison of staining intensity with acetylated tubulin antibody and Newtic1 antibody, there did not appear to be an association between the degree of acetylation of microtubules and the degree of Newtic1 expression. Nevertheless, this was useful in examining the positional relationship between microtubules and Newtic1(+) dots by confocal microscopy (Figure 3E–J). We found that many Newtic1(+) dots are assembled in close proximity to microtubules in the marginal band and are rarely localized in breaks on microtubules or in small unacetylated regions (Figure 3E,G,I).

To investigate the entity of Newtic1(+) dots, we examined PcNobs by immunoelectron microscopy. In the PcNobs where Newtic1 immunoreactivity could be observed in brown along the equator under an optical microscope, the silver-enhanced immunogold reactants were grainy and aggregated to form globular structures (typicall, 100 nm in diameter), which were aligned in close proximity to the microtubules of the marginal band (Figure 4). We examined three Newtic1(+) PcNobs and observed the same distribution pattern of globular structures. Note that under the conditions of this immunostaining, the cells’ membrane structures were not well retained due to the detergent used to enhance the permeation of the antibodies into the cells (see below). Nevertheless, importantly, the globular structures of Newtic1 did not dissociate from the microtubules.

To examine the possibility that Newtic1 is a protein that associates with microtubules, we carried out immunoblotting on protein fractions of blastemal blood cells (Figure 5). Here, two samples were prepared: sample 1, containing both water-soluble proteins in the cytoplasm and proteins solubilized with detergent from the plasma membrane, and sample 2, containing the cytoskeletal proteins, i.e., the insoluble fraction after DNA digestion (Figure 5A). We found that the Newtic1 protein was present in the cytoskeletal fraction (sample 2) as well as in sample 1 (Figure 5B). These results, together with immunocytochemical analyses (Figure 2, Figure 3 and Figure 4), suggest that Newtic1 is a protein that binds directly or indirectly to microtubules in the developing or well-developed marginal band of PcNob.

Taken together, the accumulation of Newtic1(+) globular structures just below the equator of the PcNob during limb blastema formation corresponds to the development of the marginal band, in which Newtic1(+) globular structures are tethered to its component microtubules.

### 3.2. Implication for the Newtic1(+) Globular Structure as a Membrane Vesicle Carrying Secretory Factors

#### 3.2.1. PcNobs Retain Organelles Involving Protein Packaging, Modification, and Transport

Since Newtic1 is presumed to be a membrane protein, we hypothesized that Newtic1(+) globular structures were originally membrane vesicles. In fact, it is well-known that premature erythrocytes in vertebrates have an endomembrane system [22,23]. Additionally, there is a report about nucleated erythrocytes of mammalian fetuses in which the marginal band was associated with vesicles of the ER [17]. Therefore, we investigated whether PcNobs have an endomembrane system by live cell staining (Figure 6A–C) and by TEM using cells fixed in a procedure that preserves the membrane structure (Figure 6D–F). Organelle staining of live PcNobs revealed the ER surrounding the nucleus and the Golgi apparatus (Figure 6A,B), as well as the mitochondria, although lysosomes were not evident (Figure 6C). The ER and Golgi apparatus were more developed in younger normoblasts and became limited in OcNobs (Figure 6A). TEM with ultrathin sections of PcNobs evidenced the presence of an endomembrane system, including nuclear membrane, ER, and Golgi apparatus (Figure 6D). Of note, membrane vesicles were observed in the cytoplasm of the periphery of the PcNob (Figure 6E), and small omega-shaped regions were observed in the cell membrane adjacent to the marginal band (Figure 6F), although it was inconclusive whether this indicated an endocytotic or exocytotic process. Nevertheless, these observations suggest that there is active membrane vesicle trafficking between the cytoplasm and cell membrane in PcNobs.

#### 3.2.2. Newtic1(+) Immunofluorescent Dots Are Closely Associated with TGFβ1

Based on the above experiments, it was likely that the detergent was responsible for the inability to clearly observe the plasma membrane in Newtic1 immunoelectron microscopy (Figure 4). However, it was difficult to immunostain Newtic1 protein under conditions that did not use a detergent, which dissolves the plasma membrane, ensuring antibody permeability into the cells. Furthermore, it was also difficult to immunostain Newtic1 protein directly in ultrathin sections for TEM prepared as described above (Figure 6D,F). Therefore, to obtain evidence that Newtic1 proteins are associated with membrane vesicles, here we investigated the positional relationship between Newtic1(+) immunofluorescent dots and growth factors since growth factors synthesized in the cell must often be contained in secretory membrane vesicles.

In a previous study, we showed that nearly all PcNobs in normal circulating blood express both BMP2 and TGFβ1 (Figure 7A–C and see Appendix A: [15]). In most PcNobs, BMP2 and TGFβ1 immunoreactivity was more intense in the central part of the cytoplasm (around the nucleus) and weaker at the periphery. Therefore, even in Newtic1(+) PcNobs, there was little overlap between BMP2 or TGFβ1 immunoreactivity and Newtic1 immunoreactivity, even though the host of these primary antibodies was the same (Appendix A). However, in a small number of PcNobs, TGFβ1 immunoreactivity was observed at the periphery, which, like Newtic1 immunoreactivity, was distributed in a ring along the equator (Figure 7B and see Appendix A).

In blood collected from the blastema, BMP2 immunoreactivity in the cytoplasm of PcNobs was reduced (Figure 7D). As a result, unlike intact blood, most Newtic1(+) PcNobs no longer showed BMP2 immunoreactivity. This result agreed well with the results of immunohistochemistry of blastemal sections (see Appendix A). On the other hand, the percentage of PcNobs expressing TGFβ1 along the equator significantly increased (Figure 7E; intact blood, 9.3 ± 2.8 (*n* = 3); blastemal blood, 23.3 ± 0.9 (*n* = 4); Student’s *t*-test, *p* = 0.0026). In these cells, TGFβ1 immunoreactivity in the central part of the cytoplasm decreased. TGFβ1 immunoreactivity along the equator was dot-like, similar to the characteristics of Newtic1 immunoreactivity. In double staining for Newtic1 and TGFβ1, the TGFβ1(+) dots were also stained with the secondary antibody used for Newtic1 (Figure 7G–O), though there were gaps between TGFβ1(+) granules, where only Newtic1 immunofluorescence was observed (Figure 7J,L,N; see below). Unlike intact blood, no PcNobs, whose cytoplasm along the equator was stained, ring-like, only with Newtic1 antibody, were observed. These observations were consistent with those of blastemal sections (see Appendix A).

At a higher magnification, the immunoreactive area for Newtic1 was significantly larger than that for TGFβ1 (Figure 8A): in the marginal band within a 40.9 μm^2^ image frame (100× objective lens and 10× digital zoom), TGFβ1(+) and Newtic1(+) areas were 2.3 ± 0.3 μm^2^ and 4.9 ± 0.7 μm^2^ (*n* = 6), respectively (Paired *t*-test, *p* = 0.00093). Note that in the control double stain with RFP antibody instead of Newtic1 antibody, the immunoreactive area of TGFβ1 completely overlapped with the area of the cross reaction by the secondary antibody for RFP (see Appendix A). Looking closely at the dots, it appeared as if the TGFβ1(+) dot was encompassed by Newtic1 immunofluorescence, though there were a few Newtic1(+) dots with low or no immunoreactivity to TGFβ1 (9.1 ± 0.6% in all Newtic1(+) dots, *n* = 6). Moreover, successive optical sections of Newtic1(+) dots, which were made along the Z-axis, at a minimum thickness (300 nm) and at intervals of 300 nm, also suggested that TGFβ1 immunoreactivity is present inside the Newtic1(+) dots (Figure 8B). These results suggest a possibility that the Newtic1(+) globular structure may contain TGFβ1.

Taken together, in support of our hypothesis, the Newtic1 protein may be associated with secretory vesicles that primarily carry TGFβ1 in PcNobs during blastema formation in the limb.

## 4. Discussion

As a fundamental basis for studying the physiological function of Newtic1, a membrane protein expressed in PcNobs that accumulate in the blastema of the limbs of adult *C. pyrrhogaster*, we investigated the relationship between Newtic1 and factor secretion using morphological techniques. We found that Newtic1 localizes to globular structures that accumulate in the marginal band, rather than on the cell membrane, along the equator of PcNobs, and that Newtic1 globular structures are associated with microtubules. Newtic1 immunoreactivity along the equator was found only in PcNobs with a well-developed marginal band. Considering these results with reference to the kinetics of Newtic1 expression during blastema formation in the limb [15], Newtic1(+) globular structures are thought to accumulate in the marginal bands of PcNobs, that develop in blood vessels which extend into the blastema as it grows.

PcNobs have a well-developed endomembrane system and are capable of transporting secretory vesicles from the cytoplasm to the cell membrane, and since the immunofluorescence of Newtic1 encompasses that of TGFβ1, it is reasonable to assume that Newtic1(+) globular structures are secretory vesicles. It is generally known that secretory vesicles are transported along microtubules to the cell membrane [24]. Therefore, it is hypothesized that the Newtic1 protein is involved in intracellular secretory vesicle trafficking by localizing to the membrane of secretory vesicles containing primarily TGFβ1 and binding directly or indirectly to microtubules via some proteins such as microtubule-associated proteins (MAPs) [24]. Of course, the possibility that the Newtic1(+) globular structures and TGFb1-containing secretory vesicles just happen to be in close proximity cannot be completely ruled out due to the limitations of resolution in optical microscopy. However, the fact that the single dots of Newtic1 immunoreactivity appear to encapsulate TGFb1 immunoreactivity would support this hypothesis.

It is possible that Newtic1(+) secretory vesicles carry other factors as well since there are also Newtic1(+) globular structures that are not immunoreactive for TGFβ1. However, BMP2 does not appear to be associated with Newtic1, at least during blastema growth, and another mechanism may underlie BMP2 secretion. The fact that Newtic1 immunoreactivity was not observed on the cell membrane suggests that the vesicles that fuse to the cell membrane may separate from the marginal band, leaving the Newtic1(+) membrane component behind. In PcNobs of intact blood, Newtic1(+) granules along the equator appeared to contain neither TGFβ1 nor BMP2. This may be because PcNobs have finished secreting and have returned to circulation, or there may be some other function for Newtic1(+) granules in PcNobs of intact blood.

In this study, the influence of detergent may be a concern. Note that the low density of Newtic1(+) globular structures in immunoelectron microscopy compared to confocal microscopy is not due to the detergent since the procedure up to the secondary antibody reaction was essentially the same in both techniques (see Section 2). This is primarily because the section used in immunoelectron microscopy was 80–90 nm thick, in which only one globular structure could be contained, while the optical section in confocal microscopy was at least 300 nm thick, so in areas of a high density of globular structures in the marginal band, the fluorescence of three or more globular structures would overlap.

It is likely that Newtic1 is a microtubule-associated protein and constitutes a secretory vesicle, which is why Newtic1 was able to remain tightly associated with the marginal band and TGFβ1 was able to exist in a positional association with Newtic1 even when the intracellular membrane structures were mostly lysed out by detergent. However, the exact distribution of these proteins in the endomembrane system was not visible under the current conditions. In the future, this problem should be solved by producing antibodies compatible with electron microscopy. Technology to express fluorescent-tagged Newtic1 protein in living PcNobs would enable studies of the regulation of Newtic1 expression and time-lapse analysis of the intracellular dynamics of secretory vesicles.

In mammals, including humans, TGFβ1 acts to suppress the inflammation during wound healing but also actively contributes to fibrosis and scar formation [25]. Newts, on the other hand, do not exhibit fibrosis or scar formation [1,6,9,12,13]. Our current study reiterates the possibility that TGFβ1 is secreted into the tissues during blastema formation. This suggests that in newts, TGFβ1 may contribute to scarless regeneration rather than to scar healing. Newtic1 and Newtic1(+) PcNobs appear to have a role in delivering TGFβ1, or even other factors, into tissues during blastema formation. To prove these hypotheses, further studies should be conducted in the future using genetically modified newts and in vitro experimental systems.

## 5. Conclusions

Newtic1 is a putative membrane protein expressed in premature erythrocytes and polychromatic normoblasts (PcNobs), which accumulate and secrete growth factors in the limb blastema of adult newt *Cynops pyrrhogaster* [15]. Our current results suggest that Newtic1 contributes to the secretion of growth factors, particularly TGFβ1, by localizing to the membrane of secretory vesicles, linking them to microtubules, and transporting them to the cell periphery as the marginal band develops.

## Figures and Tables

**Figure 2 biomedicines-10-02772-f002:**
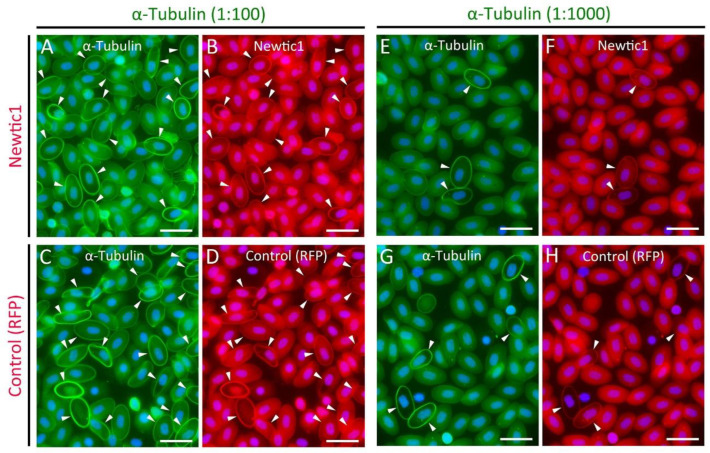
α-Tubulin and Newtic1 double stain of PcNobs in blastemal blood. (**A**,**B**) α-Tubulin (1:100) and Newtic1. (**C**,**D**) α-Tubulin (1:100) and RFP. RFP antibody was used as the control. (**E**,**F**) α-Tubulin (1:1000) and Newtic1. (**G**,**H**) α-Tubulin (1:1000) and RFP. Note that the primary antibodies used here were generated by the same host animal and that Newtic1 or RFP was stained after α-tubulin staining. Arrowheads indicate PcNobs with intense α-tubulin immunofluorescence along the marginal band. Note that the immunoreactivity of Newtic1 was not recognized in the marginal band of PcNobs, which had low or no immunoreactivity to α-tubulin (1:1000). Scale bars: 40 μm.

**Figure 3 biomedicines-10-02772-f003:**
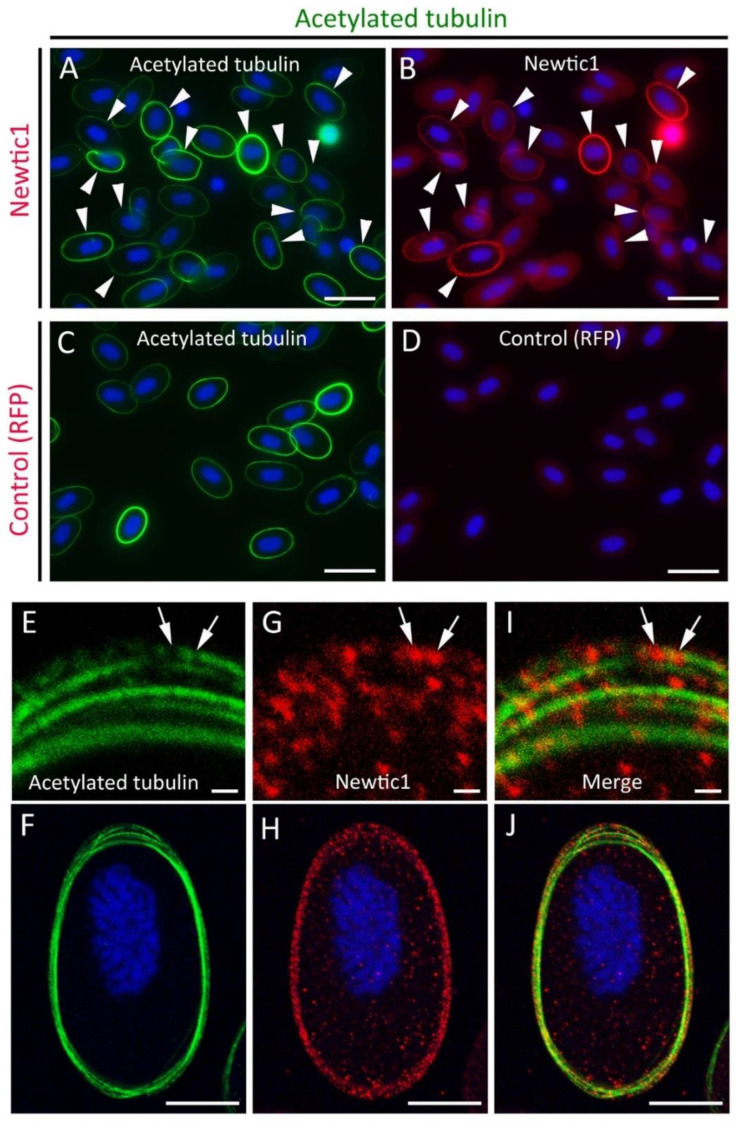
Acetylated tubulin and Newtic1 double stain of PcNobs in blastemal blood. (**A**,**B**) Acetylated tubulin and Newtic1. The acetylated tubulin antibody-stained microtubules of the marginal band at various intensities. Arrowheads indicate PcNobs with Newtic1 immunoreactivity along the marginal band. (**C**,**D**) Acetylated tubulin and RFP. RFP antibody was used as the control. (**E**–**J**) A representative set of confocal images showing the positional relationship between microtubules and Newtic1(+) dots in the marginal band (*n* = 7). (**E**,**G**,**I**) Magnified images of the upper part of the marginal band of the PcNob (**F**,**H**,**J**), where the microtubules appear to be in the process of assembling into a bundle with each other. Newtic1(+) dots were distributed in close proximity to microtubules. Arrows indicate Newtic1(+) dots, which are localized in breaks on microtubules or in small unacetylated regions. Scale bars: 40 μm (**A**–**D**); 1 μm (**E**,**G**,**I**); 10 μm (**F**,**H**,**J**).

**Figure 4 biomedicines-10-02772-f004:**
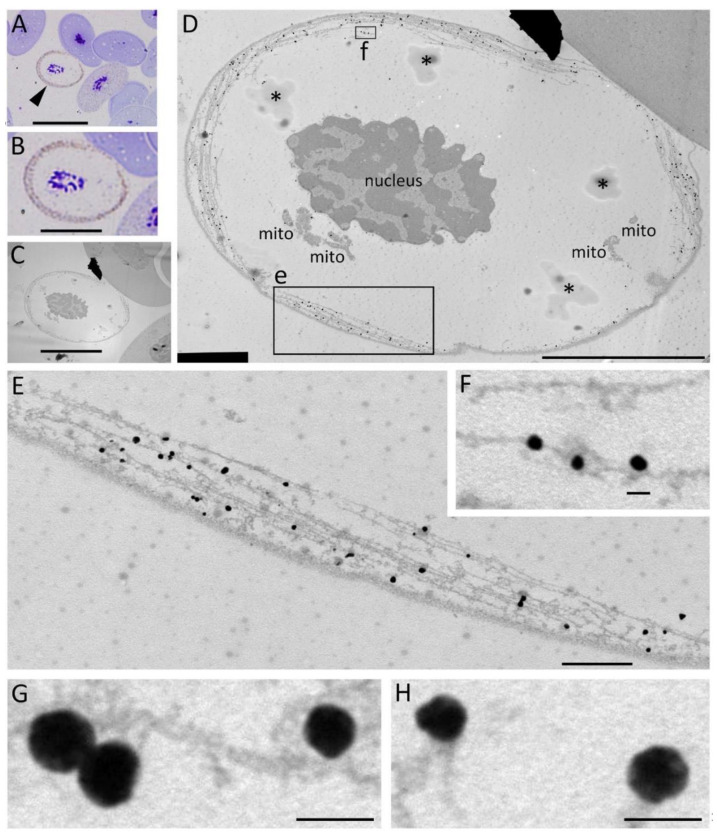
Newtic1 immunoelectron microscopy. (**A**) Optical microscopy image of Newtic1(+) PcNobs obtained with AURION R-Gent SE-LM. Cells were stained with toluidine blue. The arrowhead indicates a typical Newtic1(+) PcNob with brown dots along the equator. The cytoplasm of those cells was characteristically transparent, possibly because of less hemoglobin. (**B**) Enlargement of the cell indicated by the arrowhead in (**A**). (**C**) A transmission electron microscope image of an ultrathin section (80–90 nm thick) of the cell in (**B**). (**D**) Enlargement of the image in (**C**). mito: mitochondria. Asterisks indicate artifacts in the process of sample preparation. (**E**,**F**) Enlargement of the images in boxes e and f in (**D**). Reactants in the form of black granules were distributed along the microtubules of the marginal band. The granules were localized in close proximity to microtubules. (**G**,**H**) Magnified images of typical granules. The images were obtained from a different PcNob. The granules were globular structures with a diameter of about 100 nm, which appeared to be composed of black reactant grains. Scale bars: 40 μm (**A**); 20 μm (**B**,**C**); 10 μm (**D**); 1 μm (**E**); 100 nm (**F**–**H**).

**Figure 5 biomedicines-10-02772-f005:**
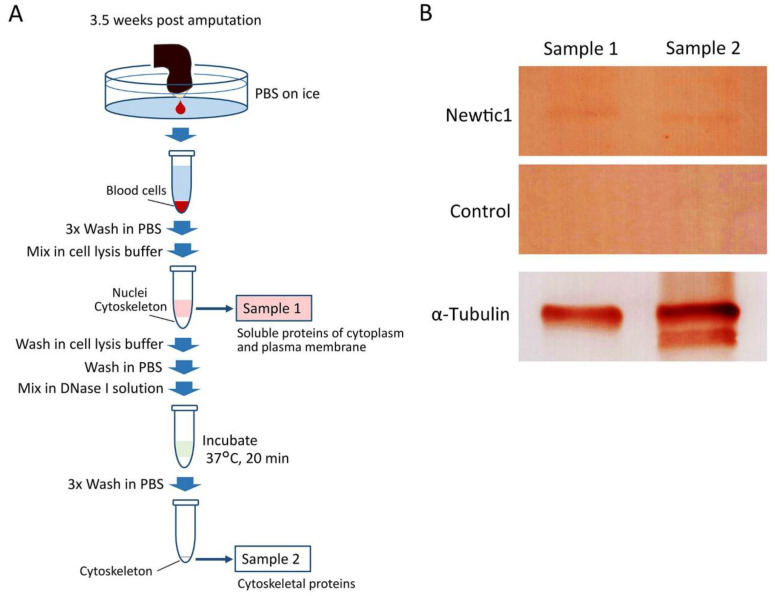
Newtic1 immunoblotting with blastemal blood cells. (**A**) Sample preparation. The proteins of blastemal blood cells were fractionated into soluble proteins in the cytoplasm and plasma membrane (sample 1) and insoluble proteins in the cytoskeleton (sample 2). For details, see Methods. (**B**) Immunoblotting. In this case, about 30 µL blood cells were collected from 5 limb blastemas (5 newts). A protein band corresponding to Newtic1 (40.7 kD) was detected not only in sample 1 but also in sample 2. Control: primary antibody omitted. α-Tubulin: 49.8 kD. For original images of the blotted membranes, see Appendix A.

**Figure 6 biomedicines-10-02772-f006:**
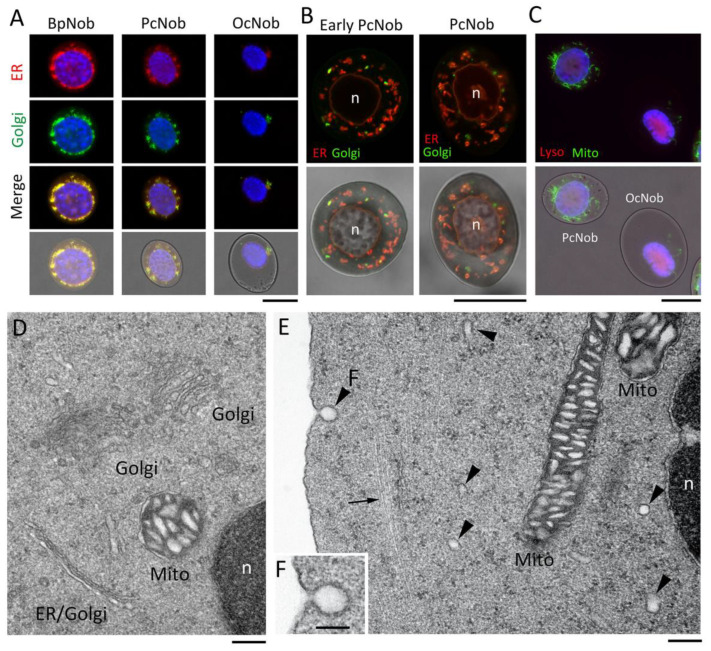
Endomembrane system in PcNobs. (**A**) Live cell staining of endoplasmic reticulum (ER), Golgi apparatus, and nucleus. Under normal fluorescence microscopy, ER (red) and Golgi apparatus (green) almost overlapped; a basophilic normoblast (BpNob) is the precursor of erythrocytes, OcNob represents mature erythrocytes, and PcNob represents erythrocytes in the transition phase. ER and Golgi apparatus were more developed in immature erythrocytes and became confined to narrow regions in OcNob. Blue: nucleus stained with Hoechst 33342. (**B**) Confocal images of live PcNobs with ER and Golgi apparatus stained. Under this condition, in which the nuclei (n) were not well stained with Hoechst 33342, the membrane structures surrounding the nucleus (i.e., nuclear membrane) were clearly visible with ER markers (red). Several Golgi apparatuses (green) were discerned as distributed at the periphery of the ER. (**C**) Live cell staining of a lysosome (Lyso), mitochondria (Mito), and nucleus. Mitochondria were, like ER/Golgi, more developed in immature erythrocytes and became confined to narrow regions in OcNob. (**D**) TEM image of Golgi apparatuses in PcNob. (**E**) TEM image of membrane vesicles in PcNob. Arrowheads indicate vesicular structures. The arrow indicates microtubules of the marginal band. (**F**) Enlargement of the membrane structure indicated by (**F**) in (**E**). An Ω-shaped membrane structure of approximately 100 nm in size was observed. Scale bars: 20 μm (**A**–**C**); 200 nm (**D**,**E**); 100 nm (**F**).

**Figure 7 biomedicines-10-02772-f007:**
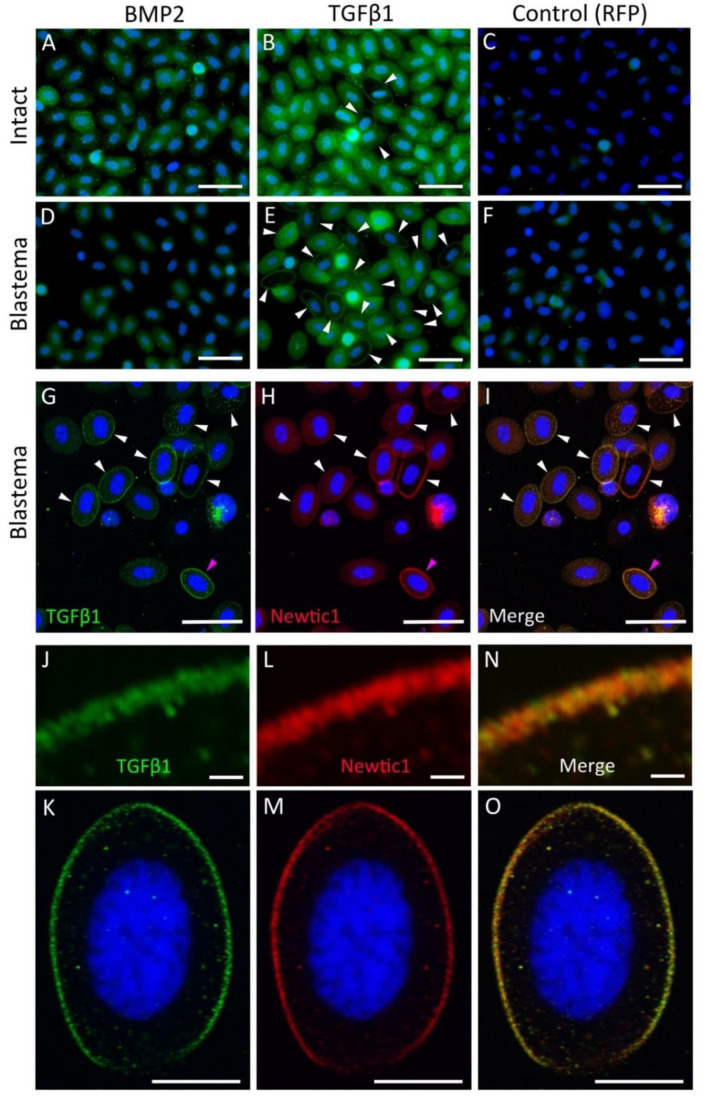
Immunoreactivity of TGFβ1 along the equator of PcNobs in blastemal blood. (**A**–**C**) Fluorescence microscopy images showing immunoreactivity of BMP2, TGFβ1, and RFP in PcNobs in intact blood. (**D**–**F**) Fluorescence microscopy images showing immunoreactivity of BMP2, TGFβ1, and RFP in PcNobs in blastemal blood. In intact blood, almost all PcNobs showed immunoreactivity to BMP2 and TGFβ1, although the intensities were variable. Both reactivities were relatively intense around the nucleus and low at the periphery (see Appendix A). However, for TGFβ1, a small number of cells also showed immunoreactivity along the equator (arrowheads in (**B**)). In blastemal blood, BMP2 immunoreactivity was reduced (**D**). For TGFβ1, there was an increase in PcNobs showing immunoreactivity along the equator (arrowheads in (**E**)). Some of these cells had decreased immunoreactivity in the cytoplasm. (**G**–**I**) A representative set of confocal images of TGFβ1 and Newtic1 double stain of PcNobs in blastemal blood. Arrowheads indicate PcNobs with TGFβ1 immunoreactivity (green) along the equator. These cells were also stained with a secondary antibody for Newtic1 immunostaining (red). (**J**–**O**) Magnified images of the PcNob indicated by the magenta arrowhead in (**G**–**I**). (**J**,**L**,**N**) Magnified images of the upper left region of the immunoreactivity along the equator of the cell shown in (**K**,**M**,**O**), respectively. Blue: nuclei stained with DAPI. Scale bars: 50 μm (**A**–**I**); 1 μm (**J**,**L,N**); 20 μm (**K**,**M**,**O**).

**Figure 8 biomedicines-10-02772-f008:**
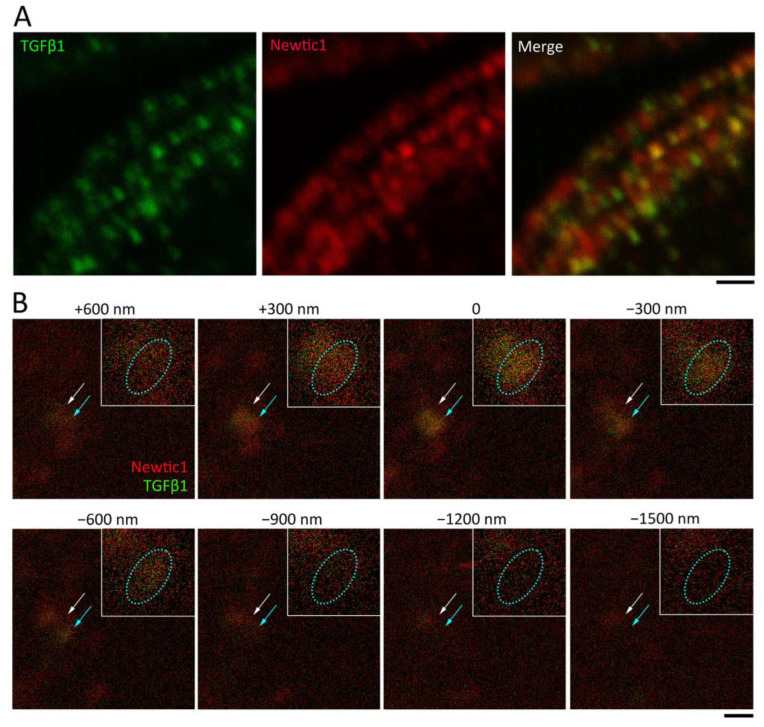
Positional relationship of TGFβ1 immunoreactivity to Newtic1 (+) dots. (**A**) Confocal images along the equator of PcNob. This cell is different from those shown in Figure 7G–I. In this cell, multiple lines of immunoreactive granules were observed just below the equator, presumably corresponding to the lines of microtubules. The immunoreactivity of TGFβ1 was granular and mostly overlapped with the region of Newtic1 immunoreactivity. However, the region of Newtic1 immunoreactivity was wider than that of TGFβ1. Note that there were also Newtic1(+) dots that did not overlap with TGFβ1 immunoreactivity. (**B**) Confocal microscopic analysis of TGFβ1(+)/Newtic1(+) dots along the Z-axis. The area with low density of dots was selected. Optical sections of maximum resolution (300 nm) were made successively at 300 nm intervals in the Z-axis direction. With respect to 0, plus indicates upward direction, and minus indicates downward direction. Arrows indicate two adjacent dots. In the dot indicated by a cyan arrow, Newtic1 fluorescence (red) was observed at +300 nm, with intense TGFβ1 fluorescence (green) overlapping with Newtic1 fluorescence at 0 and attenuated TGFβ1 fluorescence overlapping with Newtic1 fluorescence at −300 nm. The inset in each panel shows an enlarged image of the dot indicated by the cyan arrow. In the dot indicated by a white arrow, Newtic1 fluorescence was observed in the range between +600 and −300 nm, while TGFβ1 fluorescence was observed in the range between +300 and 0 nm. For the control with RFP antibody instead of Newtic1 antibody, see Appendix A. Scale bars: 1 μm (**A**); 500 nm (**B**).

## Data Availability

All data used in this study are available from the corresponding authors upon reasonable request.

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
