# Peer review of "Newtic1 Is a Component of Globular Structures That Accumulate along the Marginal Band of Erythrocytes in the Limb Blastema of Adult Newt, Cynops pyrrhogaster"

_biomedicines, 2022, doi:10.3390/biomedicines10112772_

Round 1
Reviewer 1 Report
The manuscript entitled “Newtic1 is a component of globular structures that accumulate along the marginal band of erythrocytes in the limb blastema of adult newt, Cynops pyrrhogaster” by Chen et al. deals with the investigation of the relationship between Newtic1 expression and growth factor secretion, particularly TGF-β1, by PcNobs in adult newts. Manuscript has a good concept. I appreciate the work; however, some few comments should be addressed:
Comment 1:
Although the introduction is informative, a lot of details are included especially those describing the previous study of the authors, making this section lengthy. I believe it should be shortened.
Comment 2:
Compared to confocal microscopy the dots shown by immunoelectron microscopy for Newtic1 have much less density. Although the authors mentioned that this could be resulted from the effect of detergent, can this be also explained by unspecific staining or autofluorescence on CLSM? This should be discussed.
Comment 3:
In 3.1., the authors conducted immunoblotting. The bands are somewhat hard to detect, I suggest to provide a quantification of Newtic1 protein relative to internal control. Does the ratio of cytoplasmic/skeletal proteins have a significance here?
Comment 4:
On P. 15, L. 479, it is mentioned the following: “Newtic1 is a protein that binds directly or indirectly to microtubules in the developing or well-developed marginal band of PcNob”. What do the authors mean by indirect binding of Newtic1 to microtubule? What is the suggested type of direct binding?
Comment 5:
Even though the “Conclusions” section is optional, it would be better to add it to highlights the main findings of the study.
Author Response
Response to reviewer 1:
Thank you for your comments to improve our paper.
Comment 1:
Although the introduction is informative, a lot of details are included especially those describing the previous study of the authors, making this section lengthy. I believe it should be shortened.
A1: The authors carefully discussed about the contents and length of the Introduction. As a result, hopefully, we prefer to leave it as it is. The reason is the paper's independence. That is, we have deemed this length necessary to allow readers to understand, as much as possible, the background and significance of the study as well as the complex attributes of the subject (newts, Newtic1, PcNobs, etc) by this paper alone. We hope you understand our thoughts.
Comment 2:
Compared to confocal microscopy the dots shown by immunoelectron microscopy for Newtic1 have much less density. Although the authors mentioned that this could be resulted from the effect of detergent, can this be also explained by unspecific staining or autofluorescence on CLSM? This should be discussed.
A2: The following sentences have been added to Discussion:
Note that the low density of Newtic1(+) globular structures in immunoelectron microscopy compared to confocal microscopy is not due to the detergent, since the procedure up to the secondary antibody reaction was essentially the same in both techniques (see Materials and Methods). This is primarily because the section used in immunoelectron microscopy was 80-90 nm thick, in which only one globular structure could be contained, while the optical section in confocal microscopy was at least 300 nm thick, so in areas of a high density of globular structures in the marginal band, the fluorescence of three or more globular structures would overlap.
In addition, to avoid confusion, we have added to the legend of Figure 8B, "The area with low density of dots was selected”.
Comment 3:
In 3.1., the authors conducted immunoblotting. The bands are somewhat hard to detect, I suggest to provide a quantification of Newtic1 protein relative to internal control. Does the ratio of cytoplasmic/skeletal proteins have a significance here?
A3: Thank you for the comment. In this experiment, it was difficult to increase the amount of protein, because only about 10-15 µL of blastemal blood (5-7 µL blood cells) could be collected per animal to prevent contamination of intact blood.
We have added the following sentence to 2.5 in Materials and Methods:
As for the blastemal blood, further collection should be avoided in order to minimize contamination of intact blood.
We also added the following sentence in the legend of Figure 5B:
In this case, about 30 µL blood cells were collected from 5 limb blastemas (5 newts).
Concerning quantification of Newtic1 protein relative to internal controls, it is difficult to find a suitable protein for the internal control because these samples had been fractionated. The ratio of cytoplasmic/skeletal proteins has no significance here.
Comment 4:
On P. 15, L. 479, it is mentioned the following: “Newtic1 is a protein that binds directly or indirectly to microtubules in the developing or well-developed marginal band of PcNob”. What do the authors mean by indirect binding of Newtic1 to microtubule? What is the suggested type of direct binding?
A4: Since we do not yet have a hit on the predicted function of the cytoplasmically exposed region of the Newtic1 protein, we cannot estimate whether it can bind directly to microtubules, nor can we estimate the candidate microtubule-associated proteins (MAPs) that connect it to microtubules.
So, we modified the sentence in Discussion part, instead of that in Result part, as follows:
Therefore, it is hypothesized that Newtic1 protein is involved in intracellular secretory vesicle trafficking by localizing to the membrane of secretory vesicles containing primarily TGFβ1 and binding directly or indirectly to microtubules via some proteins like microtubule-associated proteins (MAPs) [24].
Comment 5:
Even though the “Conclusions” section is optional, it would be better to add it to highlights the main findings of the study.
A5: We added Conclusions section as follows:
- Conclusions.
Newtic1 is a putative membrane protein expressed in premature erythrocytes, polychromatic normoblasts (PcNobs), which accumulate and secrete growth factors in the limb blastema of adult newt Cynops pyrrhogaster [15]. Our current results suggest that Newtic1 contributes to the secretion of growth factors, particularly TGFβ1, by localizing to the membrane of secretory vesicles, linking them to microtubules, and transporting them to the cell periphery as the marginal band develops.

Reviewer 2 Report
The authors studied the Newtic1 protein 29, which is a component of globular structures that accumulate at the marginal band in the cytoplasm along 30 the equator of PCBS. The authors showed detailed experiments and supported the explanation. Furthermore, the author provided good enough data to explain their work with reasonable discussion but there are some additional things that the authors should consider from the notes below
1. The author state on Page 3 line 118 that there is some trouble regarding regulation which is mentioned as “although newts are not specifically included in those regulations”. Regarding this sentence, if we thoroughly analyze it, is it an experiment that should not be conducted?
2. For Figure 6E, please change the small f mark to capital F.
3. The confocal images in Figure 8 are quite blurry.
4. It would be better if the author prepares an abbreviation section.
Author Response
Response to reviewer 2:
Thank you for your comments to improve our paper.
Q1: The author state on Page 3 line 118 that there is some trouble regarding regulation which is mentioned as “although newts are not specifically included in those regulations”. Regarding this sentence, if we thoroughly analyze it, is it an experiment that should not be conducted?
A1: Thank you for the comment. We explained at ‘Institutional Review Board Statement’ as follows: Ethical review and approval were waived for this study because amphibians are not included in the organisms regulated by the " Regulations for the Handling of Animal Experiments" in Japan. However, because the sentence the reviewer pointed might be a source of concern, we have modified ‘Materials and Methods’ and ‘Institutional Review Board Statement’ in the revised version as below. We have written in both the following approval number because experiments using live newts were conducted only at the University of Tsukuba, and the same animals and protocols are included in the following approved experiments at the University of Tsukuba: Approval Code: 170110 (from 1 December 2017).
- Materials and Methods
Experiments presented herein were performed at Kitasato University, Yokohama City University and University of Tsukuba. All methods were carried out in accordance with the ARRIVE guidelines as well as the Regulations for the Handling of Animal Experiments in each university. Experiments using live animals were conducted only at the University of Tsukuba. All experimental protocols for live animals were approved by the Animal Care and Use Committee of the University of Tsukuba (170110).
Institutional Review Board Statement
All experimental protocols for live animals were approved by the Animal Care and Use Committee of the University of Tsukuba (Code: 170110 from 1 December 2017).
Q2: For Figure 6E, please change the small f mark to capital F.
A2: We changed f to F in Figure 6E and legend.
Q3: The confocal images in Figure 8 are quite blurry.
A3: Thank you for the comment. As the reviewer might have suggested, the Newtic1 images in Figure 8A did not have very good contrast, and the resolution of images in Figure 8B was actually lower than the originals. We have improved these points in the revised figure. In addition, in Figure 8B, we have added an inset showing an enlarged image of the dot indicated by the cyan arrow in each panel. Accordingly, we have added an explanation in the figure legend.
Q4: It would be better if the author prepares an abbreviation section.
A4: We could not find example papers having an abbreviation section in Biomedicines. An instruction for abbreviations is as follows:
- Acronyms/Abbreviations/Initialisms should be defined the first time they appear in each of three sections: the abstract; the main text; the first figure or table. When defined for the first time, the acronym/abbreviation/initialism should be added in parentheses after the written-out form.
Alternatively, we added some words to Keywords like this:
Keywords: Newtic1, TGFβ1, BMP2, microtubule, erythrocyte, polychromatic normoblasts (PcNobs), orthochromatic normoblasts (OcNob), erythrocyte clump (EryC), limb regeneration, newt

Round 2
Reviewer 1 Report
The authors have satisfactorily addressed most of my comments.
Reviewer 2 Report
Thank you for answering my question and improving the manuscript.
I am glad to say that the revision is well-conducted, especially the ethics which are now clear.